# Compatibilizing Biodegradable Poly(lactic acid)/polybutylene adipate-co-terephthalate Blends via Reactive Graphene Oxide for Screw-Based 3D Printing

**DOI:** 10.3390/polym15193992

**Published:** 2023-10-04

**Authors:** Wei Yu, Zhonglue Hu, Ye Zhang, Yakuang Zhang, Weiping Dong, Xiping Li, Sisi Wang

**Affiliations:** 1College of Physics and Electronic Information Engineering, Zhejiang Normal University, Jinhua 321004, China; wdyuwei@zjnu.edu.cn; 2Key Laboratory of Urban Rail Transit Intelligent Operation and Maintenance Technology & Equipment of Zhejiang Province, College of Engineering, Zhejiang Normal University, Jinhua 321004, China; zhonglue.hu@zjnu.edu.cn (Z.H.); dwp@zjnu.cn (W.D.); lxp2010@zjnu.cn (X.L.); 3Beijing Aeronautical Science & Technology Research Institute (BASTRI), Commercial Aircraft Corporation of China, Shanghai 200126, China; zhangye2@comac.cc; 4Aerospace and Astronautics Propulsion Research Institute, 20 Shidai Road, Haining 314400, China; yakuang.zhang@sn-aapri.com

**Keywords:** biodegradable polymer, graphene oxide, 3D printing

## Abstract

Vinyl-functionalized graphene oxide (VGO) was used as a reactive compatibilizer to prepare poly(lactic acid)/polybutylene adipate-co-terephthalate (PLA/PBAT) blends. The linear rheological and scanning electron microscopy results confirmed that the VGO nanosheets were quite efficient in compatibilizing PLA/PBAT blends. The size of the PBAT dispersed phase was remarkably decreased in the presence of VGO nanosheets. Moreover, the VGO nanosheets exhibited strong nucleating effects on the crystallization process of PLA. The crystallinity of PLA component in the compatibilized blend with various VGO nanosheets was higher than 40%, upon the cooling rate of 20 °C/min. The prepared PLA/PBAT pellets were applied to 3D printing, using a self-developed screw-based 3D printer. The results showed that all the prepared PLA/PBAT blend pellets can be 3D printed successfully. The notched Izod impact test results showed that, in the presence of VGO, an increase of at least 142% in impact strength was achieved for PLA/PBAT blend. This could be attributed to the compatibilizing effect of the VGO nanosheets. Thus, this work provides a novel way to prepare tough PLA-based materials for 3D printing.

## 1. Introduction

Three-dimensional (3D) printing is expected to redefine consumer markets since it provides flexible and individual design methods for consumer goods [1,2,3]. In consideration of the environmental issues, biodegradable polymers are promising candidates for 3D printing consumer goods [4,5]. After their service life had ended, consumer goods made of biodegradable polymers can be decomposed simply using compost, etc. Recently, the use of biodegradable polymers in 3D printing has been successfully attempted [6,7]. Under the global concern of “white pollution” and resource exhaustion, a remarkably increasing demand for bio-based and biodegradable polymers can be expected in the field of consumer goods.

Poly(lactic acid), usually abbreviated as PLA, is known to be one of the most popular commercial biodegradable polymers. Owing to its high mechanical strength, excellent biocompatibility, and good processability [8,9], PLA has significant advantages over other biodegradable polymers. Nowadays, PLA is one of the most frequently used polymers for 3D printing of consumer goods [10]. However, the main disadvantage of PLA is its inherent brittleness [11], which significantly limits its application. Thus, elastomers and rubbers are usually blended with PLA to improve its toughness [12,13,14]. Moreover, to maintain biodegradability, which is very important here, some ductile biodegradable polymers can be blended with PLA with the aim of increasing its toughness. The most commonly used ductile biodegradable polymers are poly(butylene succinate) (PBS) [15], polycaprolactone (PCL) [16], and polybutylene adipate-co-terephthalate (PBAT) [17].

However, the toughening of PLA is quite limited when simply blending PLA with PBS, PCL, or PBAT, since they are thermodynamically immiscible with PLA [18]. Thus, interfacial compatibilization is needed to effectively prepare tough PLA-based blends. Generally, there are two main categories of interfacial compatibilization methods. One is physical compatibilizing, which enhances the miscibility of PLA through the use of a designed copolymer, which can interact with both components by forming molecular chain entanglements at the blend interfaces [19,20]. The other is chemical compatibilizing, which improves miscibility via in situ-formed compatible “copolymers” at the interface, using reactive copolymers, reactive nanoparticles, or organic peroxides [21,22,23]. The second method usually exhibits much better compatibilizing effects, producing tougher PLA-based blends. Consequently, the reactive compatibilizing technique has attracted increasing research attention in recent years.

In the pursuit of simple yet efficient reactive compatibilizers, a simple and useful vinylsilane modification technique was developed which aimed to prepare vinyl-functionalized graphene (VGN) and vinyl-functionalized carbon nanotubes (VCNTs). Moreover, both VGN and VCNTs were developed as highly efficient compatibilizers for PLA-based blends with the aid of organic peroxides [24,25]. Taking the PLA/PCL system as an example, during melt-mixing, the release of macromolecular free radicals of both PLA and PCL is initiated by organic peroxides. Then, they react with the vinyl groups of VGN to form compatibilizing “copolymers” in situ. The size of the PCL-dispersed phase was found to be remarkably decreased due to the significant compatibilizing effects. Moreover, the interfacial adhesion was largely enhanced, as evidenced by the dynamic mechanical analysis results. Consequently, the prepared PLA/PCL blends simultaneously exhibited improved tensile strength and elongation at the break.

Recent research indicated that highly reactive nanosilica with epoxy groups can effectively compatibilize PLA and PBAT [26]. This interesting work inspired the authors of the present work to explore the possibility of compatibilizing PLA and PBAT using vinyl-functionalized nanoparticles. Thus, in this work, vinyl-functionalized graphene oxides (VGOs) were synthesized and melt-mixed with PLA/PBAT with the aid of organic peroxide. The effect of the contents of the materials on the compatibilizing efficiency of PLA/PBAT blends was studied. Additionally, the compatibilized PLA/PBAT blends were reported to be suitable for fused deposition modelling (FDM) 3D printing [27]. Herein, the VGO-compatibilized PLA/PBAT blends were tested for 3D printing applications. Instead of being prepared into filaments, the VGO-compatibilized PLA/PBAT blends were directly used for 3D printing in the pellet form, using a screw-based 3D printer [28]. This kind of printer and printing technique have the advantages of lower costs, more materials available, and ease of operation. Finally, the microstructures and impact properties of the screw-based 3D-printed samples were studied. In addition, the toughening mechanism for the VGO-compatibilized PLA/PBAT blend samples was determined. 

## 2. Materials and Methods

### 2.1. Materials

Suspensions of graphene oxide (GO) with concentrations of 10 mg/mL were purchased from Suzhou Hengqiu Graphene Technology Co., Ltd., Suzhou, China. The analytically pure grade trichlorovinylsilane was purchased from Shanghai Aladdin Chemistry Co., Ltd., Shanghai, China. The PLA (Revode 190) was kindly supplied by Zhejiang Hisun Biomaterials Co., Ltd., Taizhou, China. This material had a number-average molecular weight (M_n_) of 140,000 g/mol and a melt flow rate (MFR) of 6 g/10 min (210 °C, 2.16 kg). The commercial PBAT (Ecoflex C1200) was purchased from BASF Company, Mannheim, Germany. This material had an M_n_ of 35,000 g/mol and an MFR of 4.9 g/10 min (190 °C /2.16 kg). The 2, 4-di-tert-butylisopropylbenzene peroxide (BIBP) was purchased from Shanghai RHAWN reagent Co, Ltd., Shanghai, China.

### 2.2. Preparation of Vinyl-Functionalized GO and Melt-Mixing with PLA/PBAT

The vinyl-functionalized GO (VGO) was prepared according to a technique reported in the literature [29]. In detail, the GO suspensions (100 mL) were dispersed in 400 mL of distilled water and stirred at an environment temperature of 8 °C for 0.5 h. Trichlorovinyl silane was added into the GO suspensions at a rate of 1 mL/min using an injector. Then, the dispersion was transferred into an 80 °C water bath and stirred for further reactions. After that, the obtained dispersion was centrifuged, and the reaction product was collected. The product was dispersed into distilled water again, followed by further centrifugation. This process was repeated about five times, till the dispersion was neutral. Then, the collected reaction product was dried using an air-blast-drying oven, followed by grinding via an agate mortar. The sandy powders obtained were the VGO.

The PLA and PBAT pellets were both dried in a vacuum oven for 12 h at 60 °C. Before melt-mixing, PLA, PBAT, VGO, and BIBP (dissolved in 2.5 mL ethanol) were dry-mixed sufficiently. The melt-mixing process was performed using a twin-screw extruder (HAAKE Process 11) at a screw rotation speed of 120 r/min. The temperatures were set as 130, 150, 170, 190, 190, 200, and 190 °C from hopper to die. The weight ratio of PLA to PBAT was 70/30; the weight percentages of VGO added to the PLA/PBAT compounds were set as 0.5, 1.0, 2.0, and 3.0%. The weight percentage of BIBP was fixed at 0.1% in relation to the weight of the PLA/PBAT compounds. For comparison, a PLA/PBAT/BIBP 70/30/0.1 blend was also melt-mixed with the same extrusion parameters. The as-extruded PLA/PBAT/VGO blend nanocomposites were named PVGO-x, where x indicates the weight percentage comprised by VGO.

### 2.3. Screw-Based 3D Printing

The prepared PLA/PBAT blend and PVGO blend nanocomposite pellets were formed into standard notched Izod impact samples by means of 3D printing. A self-developed screw-based 3D printer was used, as shown in Figure 1a. Differing from the traditional FDM 3D printer, the self-developed screw-based 3D printer uses a pellet-extrusion system to ensure the continual process of printing. The pellet-extrusion system works as a tiny single-screw extruder, so that the molten filaments can be continuously extruded from the nozzle for the 3D printing process. This extrusion system endows the 3D printer with obvious advantages, such as increased materials availability, since many polymers that are suitable for 3D printing may be hard to process into filaments [28].

To 3D print the impact specimens (ASTM D256 [30], Figure 1b) of both the PLA/PBAT blend and the PVGO blend nanocomposites, a nozzle with a diameter of 0.4 mm was used. The nozzle temperature was set at 220 °C, and the printing speed was 30 mm/s; the layer height was 0.2 mm, and the raster angle was 0°. 

### 2.4. Characterization

Transmission electron microscopy (TEM) observations of both the GO and VGO specimens were carried out using a TEM instrument (JEM-2100F, JEOL, Tokyo, Japan) with an accelerating voltage of 200 kV.

A rotational rheometer (MCR302, Anton Paar Instrument) was used to obtain the dynamic oscillatory shear rheological measurements. The measurements were carried out using parallel-plate geometry with a diameter of 25 mm and a gap of 1 mm. The temperature and strain were set at 220 °C and 0.5%, respectively. The scanning frequency range was set to 0.1–100 rad/s. The PLA/PBAT blend and the PVGO blend nanocomposite circular plates, made for the rheological measurements, were prepared by means of compression molding using a compression temperature of 220 °C and pressure of 10 MPa, hold-up time of 10 s, and cooling time of 1 min. 

The micro-morphologies of the PLA/PBAT blend and the PVGO blend nanocomposites were examined utilizing scanning electron microscopy (SEM). The compression-molded specimens were cryo-fractured in liquid nitrogen and were gold-sputtered for 120 s. A vacuum-coating instrument (KAS-2000F, BRIGHT) was used to carry out the gold-sputtering process and the current was set to 9 mA. After that, the specimens were examined using an SEM instrument (EM-30 plus, COXEM). The accelerating voltage was set as 15 kV, and the spot beam spot size was set to 10 nm. Moreover, to evaluate the interlayer adhesion quality of the 3D-printed samples, the morphology of the fractured surfaces on the 3D-printed samples was also examined using the SEM instrument.

Differential scanning calorimetry (DSC) measurements were performed to study both the crystallizing and melting behaviors of the PLA/PBAT blend and the PVGO nanocomposites. A DSC instrument (DSC-214, Netzsch, Selb, Germany) was applied to conduct the measurements. Specimens of about 8.0 mg were cut from the PLA/PBAT blend or PVGO nanocomposite pellets using a sharp blade. The specimens were sealed into aluminum pans, and were then scanned according to the following procedures: (1) specimens were first heated from 30 °C to 210 °C at a rate of 30 °C/min, and then maintained for 3 min to erase heat history; (2) secondly, the specimens were cooled to 30 °C at a rate of 20 °C/min; (3) lastly, the specimens were heated to 220 °C again at a rate of 20 °C/min. The endothermic peaks appearing on the DSC curves obtained at the second heating cycle were used to calculate the crystallinity (*x*_c_). In detail, the *x*_c_ was calculated according to the following equation: x_c_ = (ΔH_m_ − ΔH_c, c_)/(ΔH_f_ × C)(1)
where ΔH_m_ and ΔH_f_ are the endothermic enthalpies of PLA component and 100% crystalline PLA, respectively. A value of 93 J·g^−1^ was taken as the theoretical endothermic enthalpy of the PLA [31]. ΔH_c, c_ is the exothermic enthalpy of PLA during cold crystallization. C is the weight percentage of PLA. 

The cantilever beam (Izod type) impact test was performed using an impact tester (PTM7151, Shenzhen SUNS) equipped with a 5.5 J pendulum hammer. At least five specimens were prepared for each test for the sake of minimizing the error in measurement.

## 3. Results and Discussion

### 3.1. Morphologies of GO and VGO

Figure 2 exhibits the morphologies of both GO and VGO. Thin and aggregated GO nanosheets are observed in Figure 2a. The thinner GO nanosheets show a feature of wrinkles. This typical feature of thin GO nanosheets has also been reported in the literature [32]. For the VGO, circular black zones were observed to be decorated on the surfaces of its nanosheets (Figure 2b). These circular black zones appeared due to the presence of silicon elements, indicating a successful grafting of trichlorovinyl silane on the surface of GO nanosheets. The successful grafting of vinylsilane onto the surface of GO nanosheets has been previously evidenced using Raman spectroscopy and through X-ray photoelectron spectroscopy examines results [29]. With the intervention of vinylsilane, especially the hydrophobic vinyl groups, the aggregating tendency of VGO nanosheets is remarkably weakened. This can be evidenced by the phenomenon whereby the dried VGO films were very easy to grind into fine powders merely via a manual grinding process. This can be interpreted in the following two ways: on one hand, the hydrophobic vinyl groups largely weaken the interactions between VGO nanosheets; on the other hand, the average distances between VGO nanosheets can be expected to be increased by the intercalation of vinylsilane. 

### 3.2. Linear Rheological Behaviors of PVGO Blend Nanocomposites

Small amplitude oscillatory shear rheological measurements were performed for both PLA/PBAT blend and PVGO blend nanocomposites; the results are shown in Figure 3. As can be seen in Figure 3a,b, the dynamic storage modulus (*G*′) and loss modulus (*G*″) of both the PLA/PBAT blend and the PVGO blend nanocomposites increased with the frequency. The *G*′ curves of the PLA/PBAT blend and the PVGO blend nanocomposites exhibited a shape which is typical for polymer blends with “sea-island” morphology [33]. At low frequencies, the *G*′ is observed to obviously increase with the content of VGO. Specifically, at 0.251 rad/s, the *G*′ is increased from a value of 4.8 Pa (for PLA/PBAT blend) to 57.1 and 181.5 Pa for PVGO-2.0 and PVGO-3.0 blend nanocomposites, respectively. Such remarkable increases in *G*′ at low frequencies can be attributed to the contribution of interfacial storage modulus [34,35]. Note that the *G*′ values of PVGO-1.0 and PVGO-2.0 were quite similar in most sections of the frequency range. Only when the angular frequency was lower than 0.4 rad/s could moderately higher *G*′ values of PVGO-2.0 be observed in comparison with those of PVGO-1.0. This is a sign that VGO nanosheets cannot be well-dispersed in the PLA/PBAT blend at a certain content during the extrusion mixing process. It is plausible that, at this very content (2 wt.%), the lubricating effect of the VGO nanosheets overwhelms the thickening effect on PLA/PBAT blend melt. Thus, during extrusion mixing, the shear force cannot effectively disperse the VGO aggregates. The increased number of VGO aggregates would be expected to weaken the contribution of the interfacial storage modulus. 

The Han plot (log *G*″-log *G*′) is a useful criterion for investigating the compatibility of polymer blends. If a polymer blend is homogeneous, its Han plot usually has a slope of 2 in the terminal region [36]. The slopes of Han plots for PLA/PBAT blend and PVGO blend nanocomposites are shown in Figure 3c. In this figure, the transition zone between viscous and elastic behaviors can be seen, represented by the diagonal line. For both the PLA/PBAT blend and PVGO blend nanocomposites, the slopes are observed to deviate from the value of 2, indicating the existence of phase separation or immiscible region in the blends. However, as can be seen in the figure, the slope gradually approaches the value of 2. This indicates that the compatibility of PLA and PBAT was improved with the incorporation of VGO, and the improvement was enhanced with an increase in VGO content. 

### 3.3. Phase Morphology of PVGO Blend Nanocomposites

Figure 4 shows SEM micrographs of the cryo-fractured surfaces of the PLA/PBAT blend and the PVGO blend nanocomposites. The PLA/PBAT blend exhibits typical “sea-island” morphology, with PBAT spheres dispersed in the PLA matrix (Figure 4a). A wide distribution of the diameters of PBAT spheres can be observed due to the different merge rates of the PBAT droplets during melt-mixing. Moreover, circle-like gaps between the PBAT spheres and the PLA matrix are observed, indicating insufficient interfacial adhesion, since the two are immiscible. For the PVGO blend nanocomposites, however, circle-like gaps can hardly be observed, and the PBAT spheres are significantly reduced in size. Specifically, only tiny PBAT dots (rather than PBAT spheres) are observed, embedded in the PLA matrix for the PVGO-1.0, PVGO-2.0, and PVGO-3.0 blend nanocomposites. This shows that the miscibility and interfacial adhesion between PLA and PBAT components are largely enhanced in the PVGO blend nanocomposites. The SEM observations further confirm the linear rheological measurement results. That is, the VGO nanosheets largely improve the compatibility of PLA and PBAT.

As has been revealed in previous works, during melt-mixing, the organic peroxide initiated free radical reactions among PLA molecular chains, PCL or PE molecular chains, and vinyl groups of vinylsilane-functionalized graphene (VGN) or VCNTs [24,25]. Thus, “copolymers” of PLA and PCL, or PLA and PE linked by VGN or VCNT, formed in situ to compatibilize them significantly. Thus, much more finely dispersed phases, as well as stronger interfacial adhesion, were achieved for the blends. Herein, reactive compatibilization can be expected during melt-mixing, since the free radical reactions among PLA molecular chains, PBAT molecular chains, and vinyl groups of VGO will certainly be initiated by the BIBP. Consequently, the largely reduced phase size of PBAT can be attributed to the reactive compatibilization of the VGO nanosheets, with the aid of BIBP. 

### 3.4. Thermal Behaviors of PVGO Blend Nanocomposites

Figure 5 shows the DSC measurement results for both the PLA/PBAT blend and the PVGO blend nanocomposites. As can be seen from Figure 5a, all the samples exhibit a crystallizing peak except for the PLA/PBAT blend. The temperature of the crystallizing peak is observed to increase gradually with the content of VGO. Moreover, the onset temperature of crystallization is gradually increased from 119.9 to 125.5 °C, with the VGO content being attributable to the nucleating effect of the VGO nanosheets on the PLA component; this is because VGO nanosheets act as heterogeneous nucleating sites for the PLA crystallites. Upon heating, the crystals of the PLA component will melt again, and thus endothermic peaks can be observed. However, if the PLA component cannot crystallize well during the cooling stage, then the peak of cold crystallization will appear ahead of the endothermic peak. Herein, the peak of cold crystallization was only observed for the PLA/PBAT blend (Figure 5b). This indicated that all the samples completed their crystallization during cooling at a rate of 20 °C/min, except the PLA/PBAT blend. As can be seen in Figure 5c, the crystallinity of the PLA/PBAT blend was as low as 2.3%; whereas it sharply increased to the range of 44.9–54.1% with the incorporation of 0.5~3.0 wt.% VGO. Thus, the DSC heating curves further confirm the nucleating effect of VGO nanosheets on the PLA component. 

Usually, it is hard to crystallize PLA due to its low chain mobility. Thus, one cannot observe crystallizing peaks on the DSC cooling curves of PLA, even at a relatively low cooling rate [37,38]. The GO and graphene have been reported to exhibit nucleating effects on the PLA [39,40]. However, in this work, all four PVGO blend nanocomposites showed very high (>40%) crystallinity upon cooling. This indicates much stronger nucleating effects of the VGO nanosheets than those reported in the literature. As evidenced by both linear rheological and SEM measurement results, strong interfacial adhesion occurs in the PVGO blend nanocomposites. Thus, it is reasonable that the strong nucleating effects of the VGO on PLA are a result of the strong interfacial adhesion. On the other hand, the strong crystallizing ability of the PVGO blend nanocomposites is an obvious sign of the strong interfacial adhesion between the VGO nanosheets and molecular chains of the PLA. 

### 3.5. Microstructure and Impact Strength of 3D-Printed PVGO Blend Nanocomposites

Figure 6 exhibits the SEM micrographs of fractured surfaces on PLA/PBAT and PVGO 3D-printed samples. As can be seen, triangular holes are observed on the fractured surface of all samples. These holes are formed due to the stacking of molten filaments layer by layer during 3D printing [41,42]. The 3D-printed PVGO-2.0 sample shows more triangular holes with larger sizes than the other 3D-printed samples (Figure 6d). The height of some of the holes exceeds 100 μm. This may be ascribed to the insufficient dispersion of VGO aggregations during the extrusion of PVGO-2. Such holes with large sizes are expected to deteriorate the mechanical properties of the sample. 

Figure 7 shows the notched Izod impact strength of 3D-printed PLA/PBAT and PVGO samples. The 3D-printed PLA/PBAT samples exhibit an average impact strength of 5.7 kJ/m^2^. This is certainly higher than that of neat PLA sample, but the improvement is not large enough when one considers the weight ratio of 30% PBAT. However, at the same weight ratio of PBAT, a remarkable enhancement of the impact strength was achieved via the incorporation of the VGO nanosheets. Regardless of the content of VGO, all the 3D-printed PVGO samples exhibited an average impact strength higher than 13.8 kJ/m^2^, showing an increase of 142% for the impact strength in the presence of VGO. This can be ascribed to the compatibilization brought by the VGO nanosheets, which improves the load transfer efficiency between the PLA matrix and PBAT-dispersed phase [43]. Note that the PVGO-2.0 sample exhibited relatively lower impact strength in comparison with the other PVGO samples. This can be attributed to the higher number and larger size of triangular holes on its cross sections (as shown in Figure 6d). 

Figure 8 shows SEM micrographs of impact-fractured surfaces on PLA /PBAT and PVGO samples. Nearly no plastic deformation of the PLA matrix is observed on the impacted surface of the 3D-printed PLA/PBAT sample. Moreover, the PBAT-dispersed phases remain spherical, indicating nearly no deformation (Figure 8a). The obvious plastic deformation of both the PLA matrix and PABT-dispersed phases is observed for all four 3D-printed PVGO samples (Figure 8b–e). This further confirms that the VGO nanosheets compatibilize the PLA and PBAT components, as well as improving their interfacial adhesion. Thus, the load that occurred during impact tests can be effectively transferred from the PLA matrix to the PBAT-dispersed phases.

## 4. Conclusions

VGO nanosheets can be used as a reactive compatibilizer for PLA/PBAT blends with the aid of organic peroxides. Both linear rheological measurements and scanning electron microscopy results indicate that the VGO nanosheets were quite efficient in compatibilizing PLA/PBAT blends. Moreover, in the presence of VGO nanosheets, the crystallizing ability of PLA component was seen to be largely improved in the compatibilized blends. The prepared PLA/PBAT blend pellets were tested for 3D printing using a self-developed screw-based 3D printer. The results show that all the prepared PLA/PBAT blend pellets can be successfully 3D-printed. The notched Izod impact strength of 3D-printed PLA/PBAT blend samples can be largely enhanced by adding VGO. With the incorporation of VGO, an increase of at least 142% impact strength was achieved for the PLA/PBAT blend, thanks to its compatibilizing effect.

## Figures and Tables

**Figure 1 polymers-15-03992-f001:**
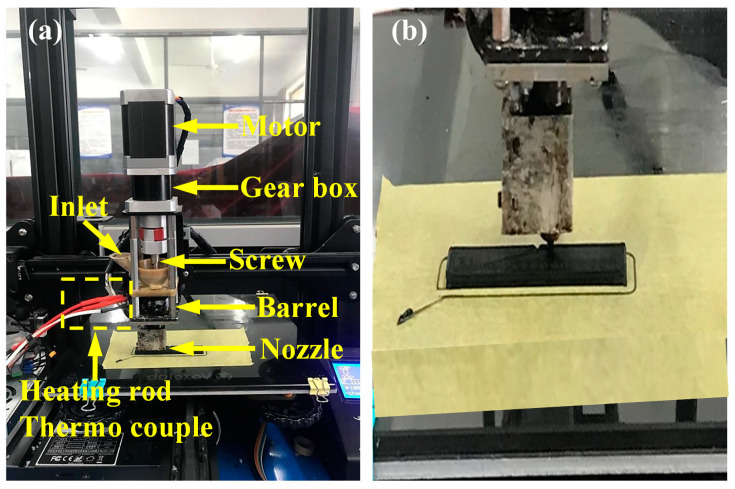
Electronic images of (**a**) self-developed 3D printer and (**b**) impact specimens.

**Figure 2 polymers-15-03992-f002:**
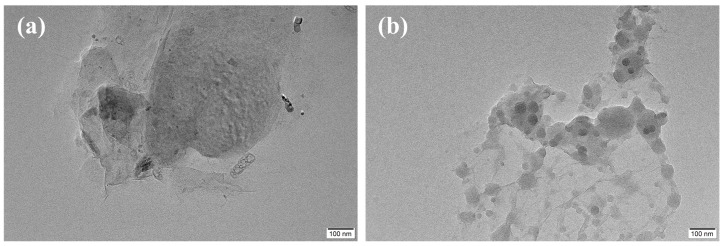
TEM micrographs of (**a**) GO and (**b**) VGO; scale bar: 100 nm.

**Figure 3 polymers-15-03992-f003:**
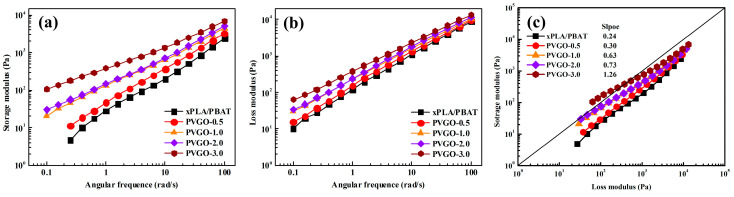
(**a**) Storage modulus, (**b**) loss modulus, and (**c**) Han plots of the PLA/PBAT and PVGO blend nanocomposites.

**Figure 4 polymers-15-03992-f004:**
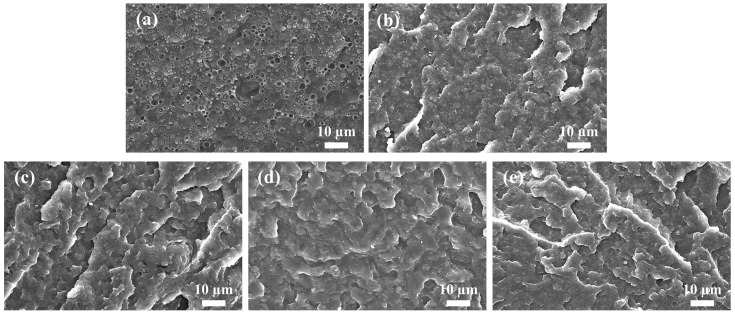
SEM micrographs of cryo-fractured surfaces of (**a**) PLA/PBAT blend, (**b**) PVGO-0.5, (**c**) PVGO-1.0, (**d**) PVGO-2.0, and (**e**) PVGO-3.0 blend nanocomposites. Scale bar: 10 μm.

**Figure 5 polymers-15-03992-f005:**
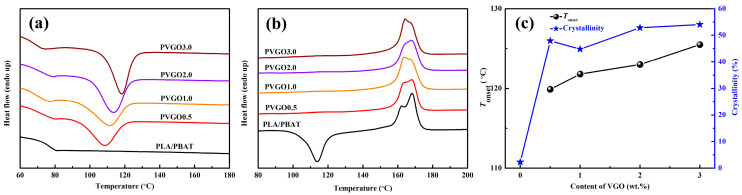
DSC (**a**) cooling curves, (**b**) heating curves, and (**c**) crystallinity, as well as the onset temperature of crystallization for PLA/PBAT and PVGO blend nanocomposites.

**Figure 6 polymers-15-03992-f006:**
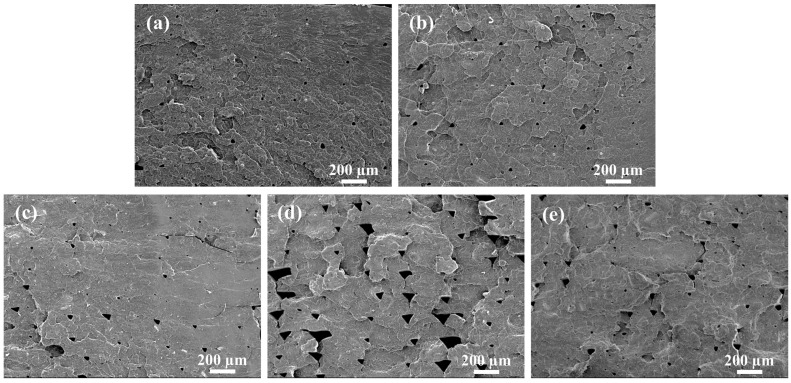
SEM micrographs of fractured surfaces on (**a**) PLA/PBAT, (**b**) PVGO-0.5, (**c**) PVGO-1.0, (**d**) PVGO-2.0, and (**e**) PVGO-3.0 3D-printed samples. Scale bar: 100 μm.

**Figure 7 polymers-15-03992-f007:**
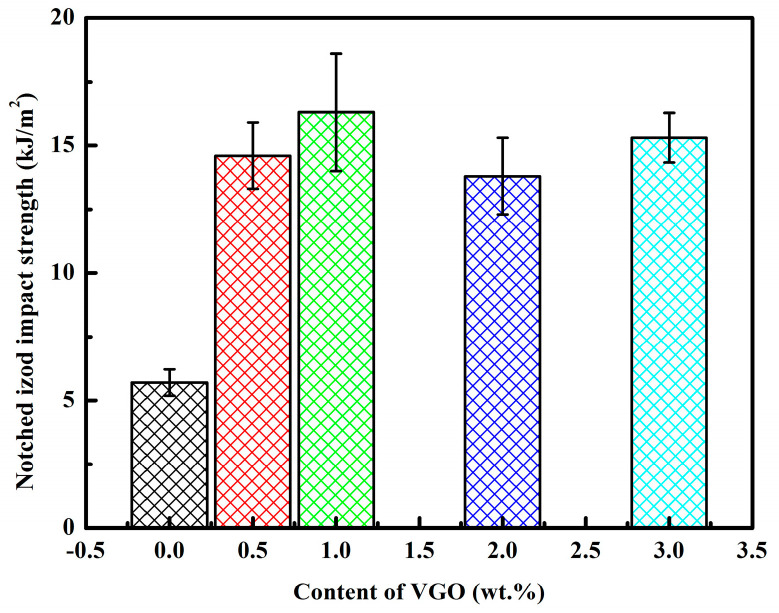
Notched Izod impact strength of 3D printed PLA/PBAT and PVGO blend nanocomposite samples.

**Figure 8 polymers-15-03992-f008:**
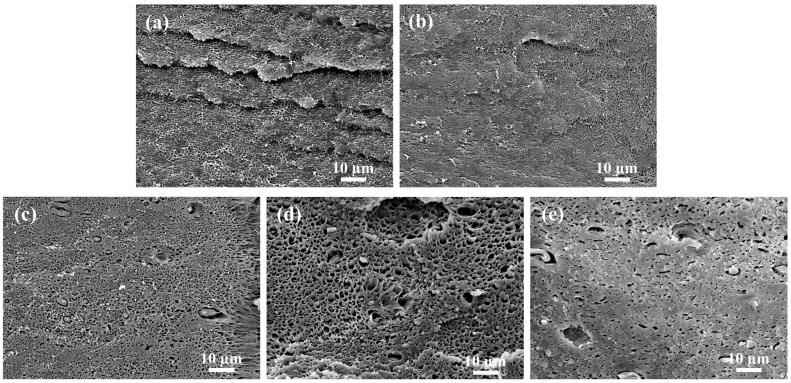
SEM micrographs of impact-fractured surfaces on (**a**) PLA/PBAT, (**b**) PVGO-0.5, (**c**) PVGO-1.0, (**d**) PVGO-2.0, and (**e**) PVGO-3.0 3D-printed samples. Scale bar: 10 μm.

## Data Availability

Not applicable.

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
