# Peer review of "Compatibilizing Biodegradable Poly(lactic acid)/polybutylene adipate-co-terephthalate Blends via Reactive Graphene Oxide for Screw-Based 3D Printing"

_polymers, 2023, doi:10.3390/polym15193992_

Round 1

Reviewer 1 Report

Generally, it is well written manuscript. However, it is not ideal, and I have only few comments.

1. I suggest to better explain eperimental procedure from lines 99-100, which wa sdifferent than in cited re. [26] (... "the reacted dispersion was centrifuged several times, followed by drying and grinding.") How it was done ?

2. An abbreviation of PBS in line 49 should be explained earlier in a text of manuscript.

3. I would be satisfied if authors could add short explanation what is advantage to use 3D processing, as compared to an ordinary extrusion of plastics.

4. It seems that similar properties of PLA/PBTA blends could be obained by using less expensive fillers than GO modified with vinyltrichlorosilane.

Author Response

Generally, it is well written manuscript. However, it is not ideal, and I have only few comments.

  1. I suggest to better explain eperimental procedure from lines 99-100, which was different than in cited re. [26] (... "the reacted dispersion was centrifuged several times, followed by drying and grinding.") How it was done ?

Thanks for the reviewer’s suggestion. This experimental procedure is rewritten in detail.

  1. An abbreviation of PBS in line 49 should be explained earlier in a text of manuscript.

Thanks for the reviewer’s kind suggestion. This is explained in the last sentence of paragraph 2 of “Introduction” section in the revised manuscript.

  1. I would be satisfied if authors could add short explanation what is advantage to use 3D processing, as compared to an ordinary extrusion of plastics.

Thanks for the reviewer’s kind suggestion. The advantage of 3D printing is that it provides more flexible and individual design, compared with injection molding and extrusion. The mold is no longer needed to form a part using a 3D printing.

  1. It seems that similar properties of PLA/PBTA blends could be obained by using less expensive fillers than GO modified with vinyltrichlorosilane.

That's true. This is due to the high compatibilizing efficiency of VGO nanosheets. At higher contents of VGO, however, aggregates may form. This will hinder the further increase of impact strength.

Reviewer 2 Report

In the introduction, it would be beneficial to mention previous works on 3D printing using PLA/PBAT materials, as well as any relevant works on filamentless/direct pellet printing techniques and the reason why you use this technique in this work. This will help engage the reader. And also mention the novelty of your work in the introduction.

In the Materials and Methods section, when discussing the 3D printing process, it would be helpful to include printing parameters such as nozzle speed, temperature, layer height, and raster angle. This information can be presented either in the text or in a table for easy reference.

Some parts of the article, such as the introduction, should be revised for consistency. So, the introduction should be written better and needs minor revisions.

Use the following resources to deepen the introduction and discussion. Shape memory performance assessment of FDM 3D printed PLA-TPU composites by Box-Behnken response surface methodology. Toughening PVC with Biocompatible PCL Softeners for Supreme Mechanical Properties, Morphology, Shape Memory Effects, and FFF Printability. 4D printing of PLA-TPU blends: effect of PLA concentration, loading mode, and programming temperature on the shape memory effect.

In the Materials and Methods section, the authors mention that they have conducted a printing process and performed a tensile test for printed parts. However, I couldn't find any corresponding section in the Results where the outcomes of these experiments are reported. It would be helpful if the authors could include a dedicated section in the Results to present the findings of these tests. This will ensure that the results are properly documented and allow for a more comprehensive understanding of the study.

In Figure 3, please provide an explanation as to why the plots converge in high frequencies?

The images presented in Figure 4 are used in raw form. Add scale bar and label for additional description.

During the DSC analysis, it would be beneficial to mention in Figure 5b that with increasing VGO content, the two peak plots transform near each other and eventually merge into one peak plot for PVGO3. This observation can serve as evidence of improved compatibility, and it would be valuable to discuss this finding.

In Figure 6, it is evident that PVGO2 exhibits decreased printability compared to the other samples. Please provide an explanation for this observation. Are there any specific reasons or factors that could contribute to this difference?

Figure 7 shows that the PVGO2 sample in the Izod test does not follow the trend observed in the other samples. Please provide an explanation for this inconsistency. It would be helpful to discuss any potential reasons or factors that could account for this deviation from the expected trend.

**

Author Response

In the introduction, it would be beneficial to mention previous works on 3D printing using PLA/PBAT materials, as well as any relevant works on filament less/direct pellet printing techniques and the reason why you use this technique in this work. This will help engage the reader. And also mention the novelty of your work in the introduction.

Thanks for the reviewer’s kind suggestion. Relevant work is cited in the revised manuscript.

In the Materials and Methods section, when discussing the 3D printing process, it would be helpful to include printing parameters such as nozzle speed, temperature, layer height, and raster angle. This information can be presented either in the text or in a table for easy reference.

Thanks for the reviewer’s kind suggestion. The above parameters are included in the revised manuscript.

Some parts of the article, such as the introduction, should be revised for consistency. So, the introduction should be written better and needs minor revisions.

Use the following resources to deepen the introduction and discussion. Shape memory performance assessment of FDM 3D printed PLA-TPU composites by Box-Behnken response surface methodology. Toughening PVC with Biocompatible PCL Softeners for Supreme Mechanical Properties, Morphology, Shape Memory Effects, and FFF Printability. 4D printing of PLA-TPU blends: effect of PLA concentration, loading mode, and programming temperature on the shape memory effect.

Thanks for the reviewer’s kind suggestion. The suggested works are cited in the revised manuscript to improve the quality if the Introduction section.

In the Materials and Methods section, the authors mention that they have conducted a printing process and performed a tensile test for printed parts. However, I couldn't find any corresponding section in the Results where the outcomes of these experiments are reported. It would be helpful if the authors could include a dedicated section in the Results to present the findings of these tests. This will ensure that the results are properly documented and allow for a more comprehensive understanding of the study.

Thanks for the reviewer’s kind suggestion. The authors really carried out the 3D printing of tensile bars. However, in this work, the authors focus on the impact strength to evaluate the toughening effect of PABT on PLA. Thus, Figure 1 and the relevant section are changed in the revised manuscript.

In Figure 3, please provide an explanation as to why the plots converge in high frequencies?

This is typical for the linear rheological tests of polymer materials. At high frequencies, all the microstructures in polymer materials cannot response well to the stimulus. Thus, the curves cannot be distinguished well at high frequencies.

The images presented in Figure 4 are used in raw form. Add scale bar and label for additional description.

Thanks for the reviewer’s kind suggestion. All the scale bars in Figure 4 is the same. The authors illustrate the length of scale bar in the figure caption of Figure 4.

During the DSC analysis, it would be beneficial to mention in Figure 5b that with increasing VGO content, the two peak plots transform near each other and eventually merge into one peak plot for PVGO3. This observation can serve as evidence of improved compatibility, and it would be valuable to discuss this finding.

Thanks for the reviewer’s kind suggestion. The two peaks on the melting curves in Figure 5b may correspond to different crystal forms of PLA component, rather than the crystals of both PLA and PBAT. Thus, the authors think this may not be suitable for evidencing the improved compatibility.

In Figure 6, it is evident that PVGO2 exhibits decreased printability compared to the other samples. Please provide an explanation for this observation. Are there any specific reasons or factors that could contribute to this difference?

 Thanks for the reviewer’s kind suggestion. This may be ascribed to the insufficient dispersion of VGO aggregations during the extrusion of PVGO-2. As can be seen from Figure 3, PVGO-1 and PVGO-2 exhibit nearly the same storage and loss modulus. This may be a signal that many VGO aggregations exist in PVGO-2.

Figure 7 shows that the PVGO2 sample in the Izod test does not follow the trend observed in the other samples. Please provide an explanation for this inconsistency. It would be helpful to discuss any potential reasons or factors that could account for this deviation from the expected trend.

 Thanks for the reviewer’s suggestion. This can be ascribed to the decreased printability of PVGO-2 sample. Higher numbers of triangular holes are observed for this sample, which will obviously decrease its impact strength.